# Accurate, scalable and integrative haplotype estimation

Olivier Delaneau [1,2]*, Jean-François Zagury[3], Matthew R. Robinson[1,2], Jonathan L. Marchini[4] &
Emmanouil T. Dermitzakis [5,6,7]

The number of human genomes being genotyped or sequenced increases exponentially and efficient haplotype estimation methods able to handle this amount of data are now required. Here we present a method, SHAPEIT4, which substantially improves upon other methods to process large genotype and high coverage sequencing datasets. It notably exhibits sub-linear running times with sample size, provides highly accurate haplotypes and allows integrating external phasing information such as large reference panels of haplotypes, collections of pre-phased variants and long sequencing reads. We provide SHAPEIT4 in an open source format and demonstrate its performance in terms of accuracy and running times on two gold standard datasets: the UK Biobank data and the Genome In A Bottle.

[1] Department of Computational Biology, University of Lausanne, Génopode, 1015 Lausanne, Switzerland. [2] Swiss Institute of Bioinformatics (SIB), University of Lausanne, Quartier Sorge – Batiment Amphipole, 1015 Lausanne, Switzerland. [3] Chaire de Bioinformatique, Laboratoire GBCM (EA7528), Conservatoire National des Arts et Métiers, HESAM Université, Paris, France. [4] Department of Statistics, University of Oxford, 24–29 St. Giles, Oxford OX1 3LB, UK. [5] Department of Genetic Medicine and Development, University of Geneva Medical School, 1 rue Michel-Servet, 1211 Geneva, Switzerland. [6] Swiss Institute of Bioinformatics (SIB), University of Geneva, 1 rue Michel-Servet, 1211 Geneva, Switzerland. [7] Institute of Genetics and Genomics in Geneva, University of Geneva Medical School, 1 rue Michel-Servet, 1211 Geneva, Switzerland. *email: olivier.delaneau@unil.ch

Haplotypes are key features of disease and population genetic analyses[1], and data sets in which they prove to be useful evolve in two main directions. On the one hand, cost reduction of single-nucleotide polymorphism (SNP) arrays allow genotyping hundreds of thousands of individuals resulting in data sets such as the UK Biobank (UKB), which regroup genotype data for half a million samples[2]. To efficiently estimate haplotypes on this scale, three methods, Eagle2[3,4], SHAPEIT3[5], and Beagle5[6,7], have been recently proposed with running times that are either linear or close to linear with sample size. Data sets consisting of millions of samples are now being generated by projects such as the Million Veteran Program[8] or by commercial companies such as 23andMe, which has now genotyped more than 5 million customers so far. In this context, it is unclear if the scaling with sample size offered by these methods is able to conveniently process genotype data on that scale. On the other hand, high-throughput sequencing now enables exhaustive screening of millions of genetic variants within tens of thousands of individuals such as in the Haplotype Reference Consortium data set[9]. Latter developments in sequencing have also witnessed the introduction of long reads technologies such as PacBio[10] or Oxford Nanopore[11]. By covering multiple nearby heterozygous variants in an individual, the long reads generated by these sequencing technologies allow resolving haplotypes across hundreds of kilobases. To achieve this task, commonly called haplotype assembly, multiple methods have been proposed so far such as WhatsHap[12], HapCut[13], or Long Ranger[14]. While these approaches are extremely efficient to resolve the phase between nearby variants, they do not allow full haplotype resolution across entire chromosome. To assemble together these blocks of phased variants, usually called phase sets, two types of approaches are explored: experimental solutions based either on Hi-C[15] or strand-seq[16] or computational solutions requiring population level data[17]. At this point, it becomes clear that haplotype estimation is now facing two main challenges: computational efficiently to accurately process large-scale data sets and data integration to exploit simultaneously large reference panels of haplotypes and long sequencing reads. In this paper, we describe and benchmark a method for haplotype estimation, SHAPEIT4, which proposes efficient solutions to these two challenges. Specifically, it allows processing either SNP array or sequencing data accurately with running times that are sub-linear with sample size, therefore making it well suited for very large-scale data sets. In addition, it also facilitates the integration of additional phasing information such as reference haplotypes, long sequencing reads, and sets of pre-phased variants altogether to boost the quality of the resulting haplotypes. To achieve this, the method builds on three main components: (i) the Li and Stephens model (LSM)[18] to capture long-range haplotype sharing between individuals, (ii) the Positional Burrows–Wheeler Transform (PBWT)[19] to speed up the computations involved in the LSM and (iii) the compact representation of the solution space built in previous versions of SHAPEIT[20,21], which allows efficient haplotype sampling and easy integration of additional phasing information[17,22]. To demonstrate its performance, we benchmark it on two gold standard data sets: the UKB[2] to evaluate its ability to process large-scale SNP array data sets and on the Genome In A Bottle (GIAB)[23] to assess its ability to leverage long sequencing read information.

## Results

### Overview of SHAPEIT4.
SHAPEIT4 improves upon previous SHAPEIT versions at two main levels. As a first major improvement, it now uses an approach based on the PBWT to quickly assemble small sets of informative haplotypes to condition on when estimating haplotypes. This provides a computationally efficient alternative to the previous approach based on

Hamming distance[20,24]. A PBWT of haplotypes is a data structure in which any two haplotypes sharing a long prefix (i.e., match) at a given position are sorted next to each other at that position. SHAPEIT4 takes advantage of this by maintaining a PBWT of all the haplotype estimates so that long matches between haplotypes can be identified in constant time. In practice, SHAPEIT4 works within overlapping genomic regions (of 2 Mb by default) and proceeds as follows to update the phase of an individual in a given region: (i) it interrogates the PBWT arrays every eight variants to get the $P$ haplotypes that share the longest prefixes with the current haplotype estimates at that position, (ii) it collapses the haplotypes identified across the entire region into a list of $K$ distinct haplotypes, and (iii) it runs the LSM conditioning on the $K$ haplotypes (Fig. 1a, b). In this approach, $P$ is the main parameter controlling the trade-off between speed and accuracy and gives a model in which $K$ varies and adapts to the data and region being processed as opposed to previous methods in which $K$ is usually fixed[20,24]. Indeed, $K$ varies depending on the length of the matches found in the PBWT: longer matches involve smaller $K$, which typically occurs as the algorithm converges, as the level of relatedness between individuals increases and more importantly as the number of samples in the data set increases. The latter implies, and we later show, that SHAPEIT4 scales sub-linearly with sample size. All other methods proposed so far exhibit at best linear or close-to-linear scaling. In other words, the time spent per genome decreases as the total number of genomes being processed increases.

As a second improvement, SHAPEIT4 offers the possibility to integrate three additional layers of phasing information when estimating haplotypes: a reference panel of haplotypes, phase information contained in sequencing reads and subsets of genotypes at which the phase is known a priori (termed as haplotype scaffold). This builds on previous work developed as part of SHAPEIT2[17,22] so that all these layers of information can be conveniently and simultaneously used. Adapting the method to leverage reference panels of haplotypes is straightforward: the PBWT matching procedure was just extended to also consider the reference haplotypes when selecting conditioning haplotypes. Concerning the two other layers of information, we implemented them as constraints in the haplotype sampling scheme. Specifically, we leverage phase information contained in sequencing reads in a two-step approach in order to easily accommodate with new sequencing technologies as they are developed. First, we perform haplotype assembly with methods such as WhatsHap for instance[12]. This essentially regroups nearby heterozygous genotypes into phase sets when they are overlapped by the same sequencing reads (Fig. 1c). Then, we model the resulting phase sets as probabilistic constraints in the SHAPEIT4 haplotype sampling scheme so that haplotype configurations consistent with them are favored but not necessarily sampled. This is controlled by a parameter that defines the expected error rates in the phase sets (default is 0.0001). As a consequence, depending on the certainty of the population-based phasing calls, we basically have two possible scenarios: (i) uncertain calls can get informed by the phase sets, which typically occur at rare variants and (ii) calls with high certainty can correct phase sets when they contain errors. For the haplotype scaffold, we explored two possible strategies in this work: a family-based scaffold that we derived from genotyped parents and a population-based scaffold that we derived from very large reference panels of haplotypes (Fig. 1d). In both cases, this gives reliable haplotype estimates defined at a sparser set of variants that we leverage by enforcing SHAPEIT4 to only sample haplotypes that are fully consistent with the available haplotype scaffolds. This helps the algorithm to converge towards good resolutions by pruning out unlikely configurations.

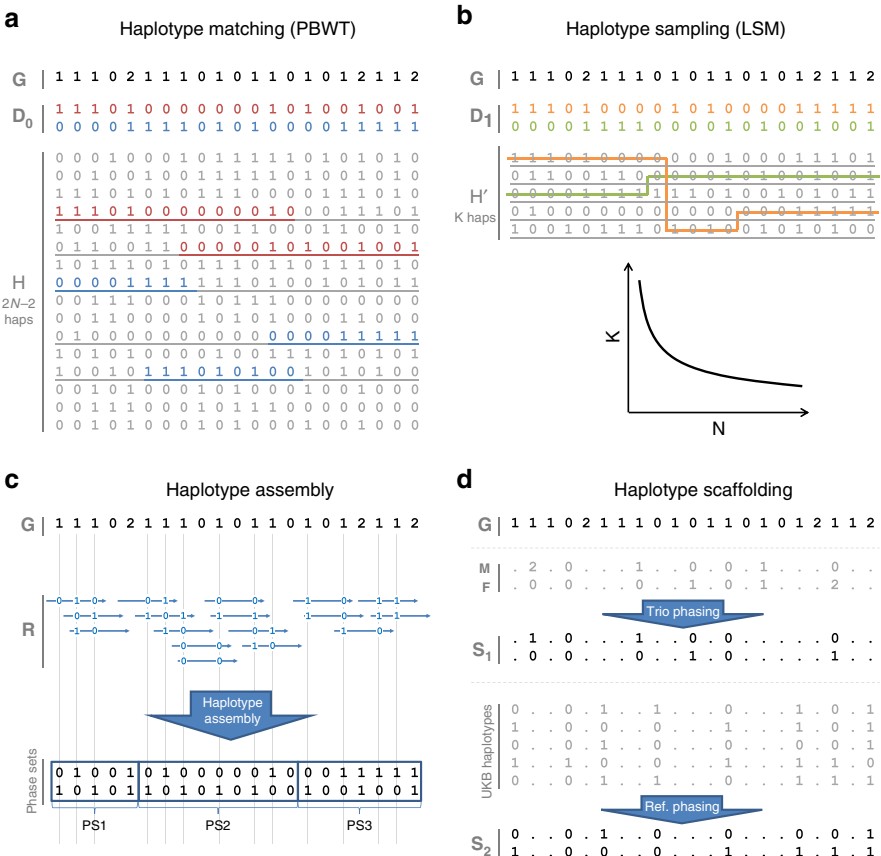

**Fig. 1** SHAPEIT4 overview. In all panels is shown the unphased genotype data for individual **G**. **a** Selection of a small number of informative haplotypes. Conditioning haplotypes with long matches with the current estimate $D_0$ identified by PBWT are underlined. Matches are shown in red and blue depending on the matched haplotype. **b** Illustration of the Li and Stephens model run on the five informative haplotypes with long matches identified with PBWT. This gives new estimates $D_1$ for **G**. **c** An example of a set **R** of phase informative reads (i.e., overlapping multiple heterozygous genotypes) for individual **G**. Haplotype assembly on **R** gives three phase sets (PS1, PS2, and PS3). The phase sets are the information used by SHAPEIT4 when estimating the haplotypes for **G**. **d** Two examples of haplotype scaffolds for **G**. $S_1$ is derived from trio information (**M** and **F** are genotype data for the mother and father of **G**). $S_2$ is derived from a reference panel such as UKB. Only variants in **G** in the overlap with UKB are phased using all UKB haplotypes as reference panel. Source data are provided as a Source Data file.

**Phasing large-scale data sets**. We assessed the performance of SHAPEIT4 on genotype data sets with large sample sizes and compared it to the following methods: SHAPEIT3[5], Eagle2[3,4], and Beagle5[6,7]. To do so, we built multiple subsets of the UKB data set comprising up to ~400,000 individuals, 500 of them being trio children for whom haplotypes can be derived with certainty from the family information. We used data on chromosome 20, which comprises 18,477 SNPs. For each phasing scenario, we measured the overall running times and the switch error rates (SERs) on the 500 trio children. Overall, we found that all tested methods provide haplotype estimates with low error rates, which substantially decrease as sample size increases (Fig. 2a). A closer look reveals that both Beagle5 and SHAPEIT4-P = 4 significantly outperforms all other methods across all tested sample sizes. For instance, on the largest sample size, we get the following error rates: SHAPEIT4-P = 4 (0.117%), Beagle5 (0.125%), SHAPEIT4-P = 2 (0.139%), Eagle2 (0.178%), SHAPEIT4-P = 1 (0.202%), and SHAPEIT3 (0.356%). As expected, increasing P yields to appreciable improvements in accuracy: for instance, the P = 2 and P = 4 configurations decrease the error rate of P = 1 by 31 and 42% on the largest sample size, respectively. This shows that having multiple candidates to copy from at a given position helps the algorithm to reach good estimates. Accuracy of the haplotype estimates can also be assessed by looking at the mean length of haplotype segments free of any switch errors. These segments

become very long when phasing 400,000 samples: 15.50 and 14.75 Mb for SHAPEIT4-P = 4 and Beagle5, respectively (Supplementary Fig. 1). In terms of running times, we found SHAPEIT4 to be substantially faster than all other methods regardless of the sample size (Fig. 2b). For instance, SHAPEIT4-P = 4 is 1.6 to 3.6 times faster than Beagle5, 1.9 to 5.8 times faster than Eagle2, and 4.1 to 11 times faster than SHAPEIT3 when phasing from 10,000 to 400,000 individuals. The speedup gets better with sample size as a consequence of sub-linear scaling. Indeed, SHAPEIT4 spends less time per genome on larger sample sizes conversely to all other methods (Fig. 2c). The sub-linear scaling can also be noted when looking at the variation of the number of conditioning states with sample sizes and iterations (Supplementary Fig. 2A–C) or when looking at the coefficients of the fitted functions relating running times T to sample size N on this benchmark data set (Supplementary Fig. 2D–F). This allows SHAPEIT4 to offer the best trade-off between speed and accuracy across all tested methods (Fig. 2d). To assess its robustness in other situations, we then run it on two additional data sets: one based on UKB in which we introduced variable amounts of genotyping errors and another based on 1000 Genomes samples that mixes samples from different ancestries. In both cases, SHAPEIT4 performs well (supplementary Figs. 3 and 4) demonstrating that the PBWT-based approach adapts well in such situations owing to its ability to select a variable number of

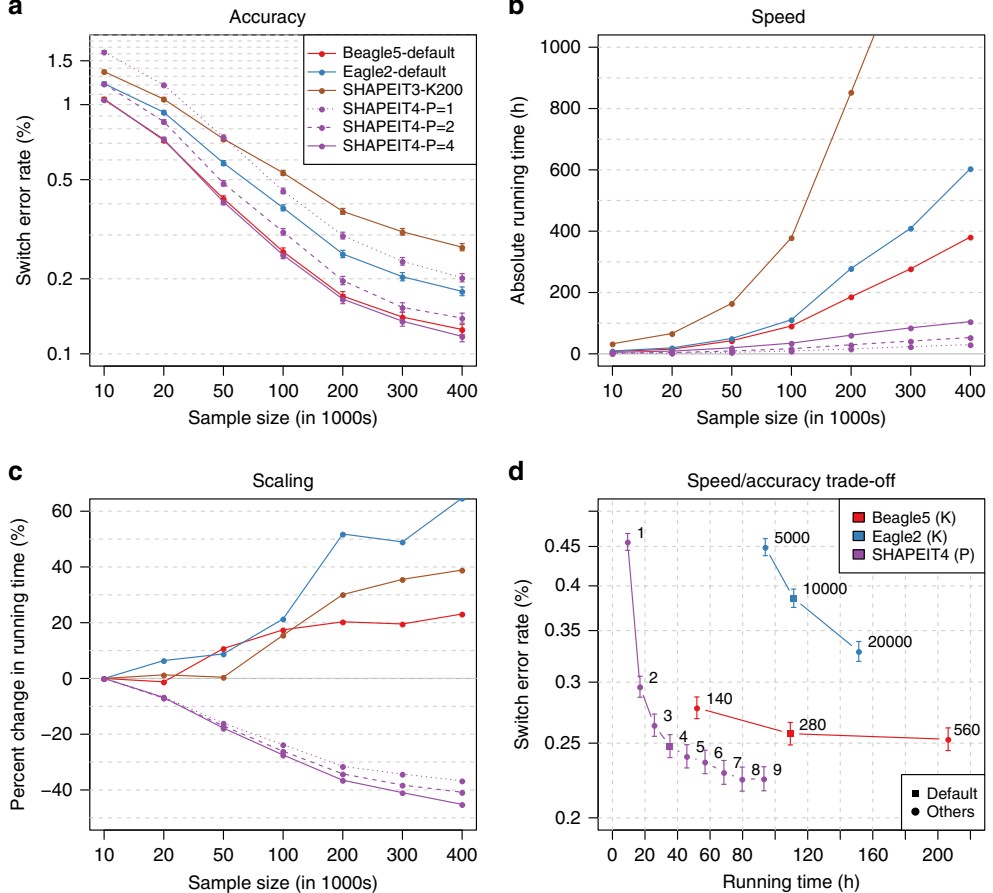

**Fig. 2** Phasing performance on large sample sizes (UKB). **a** Switch error rates for all tested phasing methods as a function of sample size (going from 10,000 to 400,000 individuals). For each error rate is shown the 95% binomial confidence interval. **b** Corresponding running times in hours measured using LINUX time command (User + System time). **c** Percentage change in running time needed to phase a single genome out of 10,000 to 400,000 genomes relative to the time needed to phase one genome out of 10,000. Positive, null, or negative slopes are indicative of close-to-linear, linear or sublinear scaling, respectively. **d** Running time as a function of switch error rates for various parameter values controlling the accuracy of the tested methods: the fixed number of conditioning states $K$ in case of Beagle5/Eagle2 and the number $P$ of PBWT neighbors for SHAPEIT4. This was run for 100,000 UKB genomes. 95% Binomial confidence interval are given. Source data are provided as a Source Data file.

conditioning states. Now, in terms of memory usage, we also found SHAPEIT4 to substantially improve upon SHAPEIT3 (3.5× decrease) and to have memory requirements that match available hardware capabilities as Beagle5 and Eagle2 do (Supplementary Fig. 5). For instance, on the largest sample size, we get the following ranking: Eagle2 (8.8 Gb), SHAPEIT4-P = 1 (30.6 Gb), SHAPEIT4-P = 2 (37.9 Gb), Beagle5 (47.3 Gb), SHAPEIT4-P = 4 (52.3 Gb), and SHAPEIT3 (182.4 Gb). Overall, we found in this first benchmark that SHAPEIT4 offers the best compromise between accuracy, speed, and memory usage across all tested methods.

**Phasing from large reference panels**. We also assessed the performance of SHAPEIT4 when phasing from large reference panels of haplotypes. To do so, we used the UKB haplotypes exhibiting the smallest SERs in the first benchmark in which we removed the haplotypes of the 500 trio children. This resulted in reference panels containing from ~1000 to ~800,000 reference haplotypes. We then phased against these candidate reference panels the 500 trio children alone or as part of larger data sets comprising 5000 to 50,000 individuals, using either SHAPEIT4, Beagle5, or Eagle2. For each phasing scenario, we measured the overall running times and the SERs on the 500 trio children. In this second benchmark, we found the same accuracy patterns

than in the first benchmark: overall error rates decrease as reference panel size increases and both SHAPEIT4-P = 4 and Beagle5 outperform all other methods no matter the amount of data being processed (Fig. 3a–c). In terms of running times, we found SHAPEIT4 to be faster than any other methods in all scenarios excepted one (Fig. 3d–f): when phasing few samples (=500) from very large reference panels (>400,000 haplotypes) where Eagle2 seems to be slightly faster (Fig. 3d). Interestingly, we also found that the PBWT approach used by SHAPEIT4 to have an interesting property in this particular context: phasing large sample size sizes (=50,000 samples) is faster when using large reference panels (Fig. 3f). Again here, we found SHAPEIT4 to provide overall the best trade-off between speed and accuracy when phasing from large reference panels.

**Phasing using sequence reads**. We finally assessed the ability of SHAPEIT4 to leverage additional phase information when processing genotypes derived from high-coverage sequencing data. To do so, we merged the GIAB high-coverage genotype data on chromosome 20 ($n = 1$) with unrelated European individuals that have been sequenced as a part of the 1000 Genomes project[25] ($n = 502$), resulting in a data set comprising 503 individuals typed at 507,181 variants. We then phased this data set using SHA-PEIT4 and two different layers of phase information. First, we

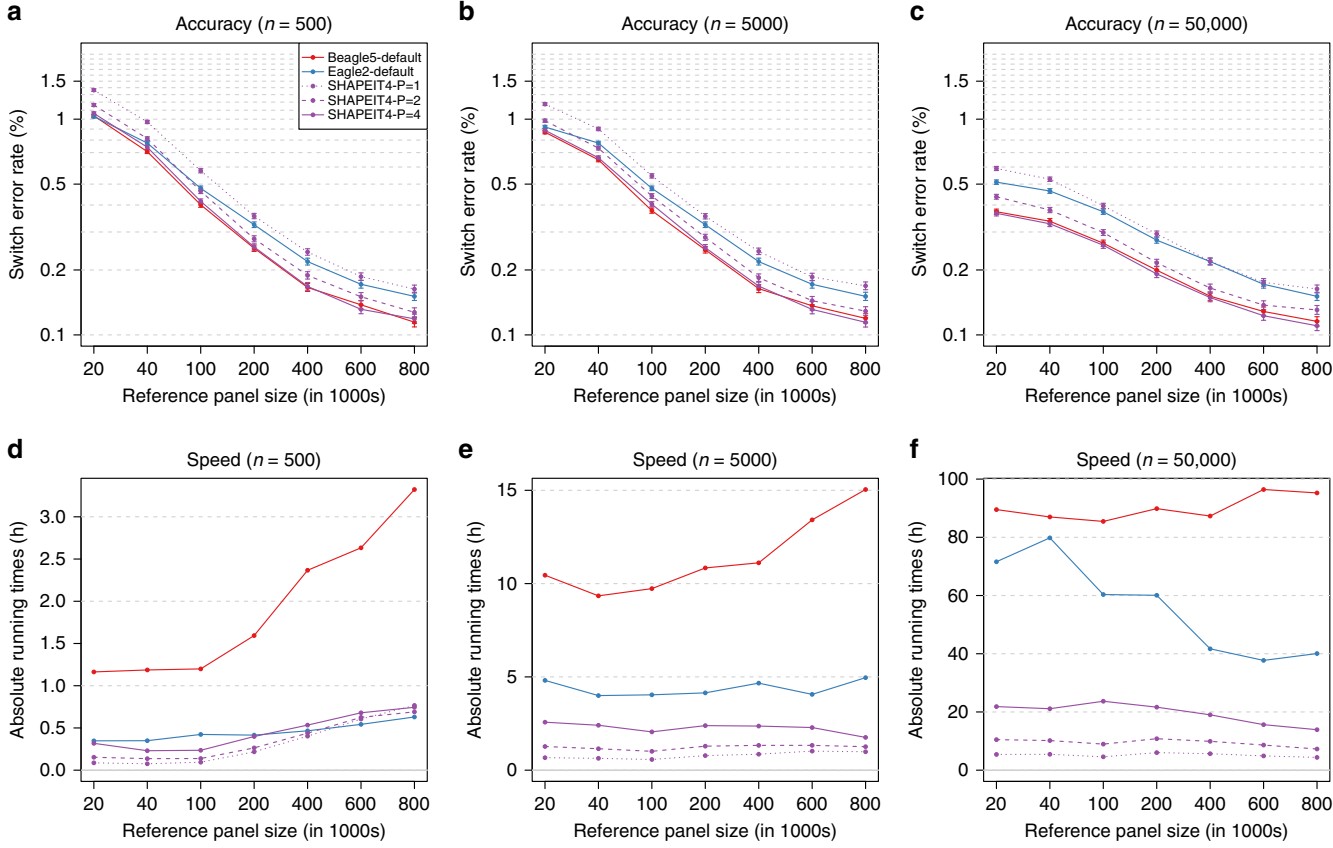

**Fig. 3** Phasing performance from large reference panels (UKB). Switch error rates (**a–c**) and running times (**d–f**) for all tested phasing methods as a function of the number of haplotypes in the reference panels. Three different sample sizes were tested for the main panel: 500 (**a**, **d**), 5000 (**b**, **e**), and 50,000 (**c**, **f**). For each error rate is given the 95% binomial confidence interval. Source data are provided as a Source Data file.

**Table 1 Summary statistics for GIAB sequencing data.**

|  | No. of reads (millions) | Mean read length (bp) | Mean insert size (bp) | Mean coverage | %hets phased |
|---|---|---|---|---|---|
| Solid5500W | 22.13 | 68 | NA | 22 | 9.8 |
| Complete Genomics | 132.6 | 33 | 282 | 77 | 35.1 |
| Illumina HiSeq | 13.29 | 148 | 546 | 33 | 57.9 |
| PacBio | 1.04 | 3707 | 2048 | 42 | 90.1 |
| 10x Genomics | 22.27 | 88 | 194 | 33 | 97.7 |

For each type of sequencing data available for GIAB is given the total number of reads in millions, the mean length of the reads in base pairs, the mean insert size in base pairs, the means coverage, and the percentage of heterozygous genotypes belonging to phase sets

performed haplotype assembly (using WhatsHap; Fig. 1c) for the GIAB genotype data using five types of sequencing data[23]: SOLiD5500W, Complete Genomics, Illumina HiSeq, PacBio, and 10x Genomics, which informed the phase at 9.8%, 35.1%, 57.9%, 90.1%, and 97.7% of the heterozygous genotypes, respectively (Table 1). Second, we built two different haplotype scaffolds for this data set: (i) one derived from a larger set of samples genotyped on Illumina OMNI2.5M SNP arrays in which most heterozygous genotypes can be phased using duos/trios and (ii) another one derived by phasing all 503 samples against all available UKB haplotypes obtained in the first benchmark (~800,000 haplotypes; Fig. 1d). The first scaffold covers 197 (=39%) individuals (including NA12878) at 43,176 variants (=8.5%) and fixes the phase of 6.59% of all the heterozygous genotypes. The second scaffold covers all 503 individuals at 16,805 variants (=3.3%) and fixes the phase at 6.25% of all the heterozygous genotypes. In this third benchmark, we could make the following observations. First, the error rates of

SHAPEIT4 significantly decrease when using the phase information contained in the phase sets, demonstrating the ability of the methods to leverage such information (Fig. 4a). Second, the various sequencing technologies exhibit large differences in terms of accuracy. Not surprisingly, technologies based on long or barcoded reads produce more accurate haplotypes with error rates as low as 0.23% and 0.07% for PacBio and 10x Genomics, respectively (Fig. 4a). Third, sparse haplotype scaffolds derived either from family information or large reference panels provide appreciable boosts in accuracy, particularly at common variants (minor allele frequency >1%; Fig. 4b). This demonstrates the benefit of using haplotype scaffolds when phasing sequencing data. Fourth, SHAPEIT4 allows correcting many switch errors introduced at the haplotype assembly step (Fig. 4c). For instance, in our data, it corrects 26% and 79% of the switch errors obtained from Illumina HiSeq and PacBio, respectively. Fifth, the error model for phase sets performs well regardless of the expected error rates (Supplementary Fig. 6) and at least as well as the

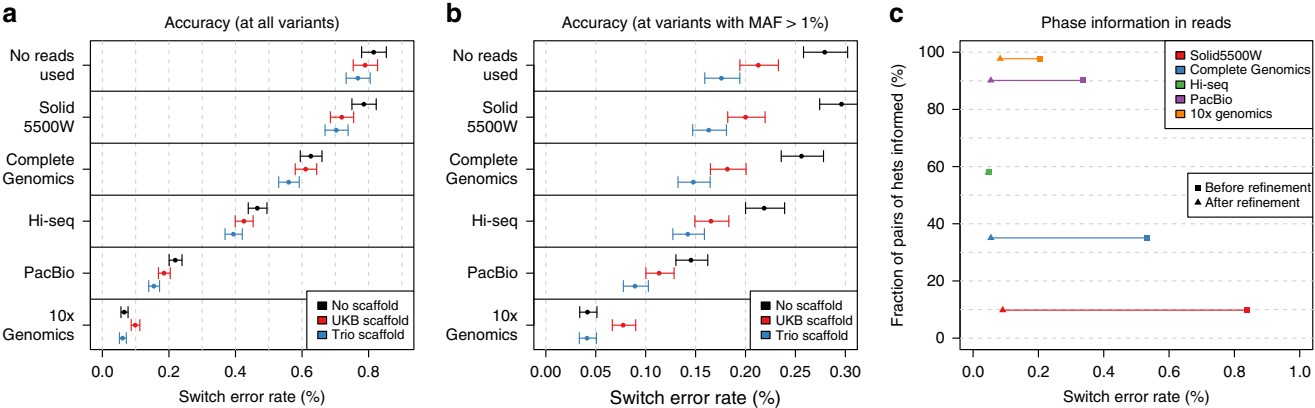

**Fig. 4** Phasing performance on high-coverage sequence data (GIAB). **a** Switch error rates with 95% binomial confidence intervals for each combination tested of sequencing reads and haplotype scaffold. **b** Same information than in **a** measured only at variants with minor allele frequency (MAF) above 1%. **c** Switch error rates measured only at variants belonging to phase sets (i.e., haplotype assembly) before (squares) and after (triangles) refinement by SHAPEIT4. Results are shown here assuming 0.01% error rate in the phase sets. Source data are provided as a Source Data file.

model used in SHAPEIT2 while being able to process any type of sequencing data (Supplementary Fig. 7). Finally, the value of integrating data using SHAPEIT4 can be summarized by comparing three phasing scenarios of interest (Supplementary Fig. 8): (i) the default situation in which the data was directly phased without using any additional information (SER = 0.82%), (ii) the situation that can often be implemented in which both Illumina HiSeq data and a UKB based haplotype scaffold are used in the estimation (SER = 0.42%, 49% decrease in error rate), and (iii) the most accurate situation in which both 10x Genomics data and a trio based haplotype scaffold are used (SER = 0.07%, 91% decrease in error rate).

## Discussion

We present here a method for statistical haplotype estimation, SHAPEIT4, that substantially improves upon existing methods in terms of flexibility and computational efficiency. One of the key improvements resides in its ability to quickly select a variable number of informative haplotypes to condition on through a PBWT-based approach. The resulting method exhibits sub-linear running times with sample size and provides highly accurate haplotype estimates for large-scale data sets. We demonstrated this on the largest genotype data set available so far, the UKB, in which SHAPEIT4 offers the best trade-off between speed and accuracy across all tested methods. We predict SHAPEIT4 to be particularly well suited to process data sets reaching a million individuals and above. Of note, the use of PBWT in the context of phasing was pioneered in Eagle2[3,4] in order to build compact representations of a fixed number of conditioning haplotypes (called HapHedge structures). This fundamentally differs from our approach in which we use PBWT to identify a small and variable number of informative haplotypes to condition on. In addition, even though our use of PBWT in this context involves overall running times that increase sub-linearly with sample size, this does not formally mean that the SHAPEIT4 algorithm has sub-linear complexity as it still contains two components with linear complexity: I/O operations and PBWT passes. However, these two linear components are expected to have small contributions to the overall running times in comparison of the HMM-based computations for sample sizes in the order of millions. Beyond phasing, we believe this PBWT-based approach to have the potential to speed up computations involved by other haplotype-based models used in population genetics for admixture mapping[26], identity-by-decent (IBD) mapping[27] or genotype imputation[24,28]. Nowadays, haplotype phasing is commonly

performed on imputation servers such as the UMich server based on large reference panels of haplotypes. We demonstrated that the initial implementation of SHAPEIT4 already provides high performance in this context and we predict that this can be further improved by using other PBWT algorithms (e.g., Algorithm 5 from Durbin[19]). Besides computational efficiency, SHAPEIT4 is also flexible enough to conveniently leverage additional phase information contained in long sequencing reads. We notably showed that SHAPEIT4 is able to deliver very accurate haplotypes for high-coverage sequenced data when leveraging long or barcoded reads through a first step of haplotype assembly (e.g., 10x Genomics). In addition, its ability to use haplotype scaffolds offers an interesting framework in which the high accuracy delivered by very large SNP array data sets can be propagated into high-coverage sequencing data sets of smaller sample sizes. We predict that these two functionalities will help improving phasing servers dedicated to whole-genome sequenced samples[29]. It is also important to mention here that SHAPEIT4 is not designed to perform genotype calling from low coverage sequencing and adapting the model for such data would need some additional work. Overall, we believe SHAPEIT4 to be particularly well suited to fully exploit the potential of the large data sets being generated using either SNP arrays or new sequencing technologies.

## Methods

**Haplotype sampling**. Let us assume we have genotype data for $N$ individuals across $L$ variant sites that we want to phase in $2N$ haplotypes. To achieve this, SHAPEIT4 uses an iterative approach. It takes each individual's genotype data $\mathbf{G}$ in turn and estimates a consistent pair of haplotypes $\mathbf{D} = (h_1, h_2)$ given the set $\mathbf{H}$ of haplotypes that has been previously estimated for the other $N - 1$ individuals. When repeated many times, this builds a Monte Carlo Markov chain (MCMC) is in which a single iteration consists in updating the haplotypes of all $N$ individuals. To update the haplotypes of $\mathbf{G}$, SHAPEIT4 proceeds probabilistically by sampling a new pair of haplotypes from the posterior distribution P(D|H). This posterior is based on a particular Hidden Markov Model, the LSM, that build haplotypes for $\mathbf{G}$ as mosaics of other haplotypes in $\mathbf{H}$. This builds on the idea that individuals in a population share relatively long stretches of haplotypes inherited from common ancestors. While being accurate, the LSM is also very computationally demanding and rapidly becomes intractable on large data sets. To ameliorate this, we introduced in SHAPEIT1 an algorithm able to sample from P(D|H) in linear times with the size of $\mathbf{H}$. We achieved this by carrying out all LSM computations on compact graph representations of all possible haplotype configurations that are consistent with each individual (called genotype graph; see Supplementary Fig. 9 for a graphical description and Delaneau et al.[21] for a formal one). In SHAPEIT4, we essentially use the same sampling algorithm that we completely re-implemented for better performance:

- We used bitwise arithmetic for all genotype graph construction and manipulation routines which resulted in ~100× speed-ups in some cases.

- We defined more compact data structures to store genotype graphs in memory, which resulted in a >3× decrease in memory usage.
- We optimized the code for HMM computations to make it cache friendly and vectorised. This speeded up the HMM computations by ~3×.

**Haplotype selection**. A common approach to further speed up the sampling from P(D|H) is to only use a subset of $K$ conditioning haplotypes instead of the full available set of $2N-2$ haplotypes (with $K \ll 2N-2$). In SHAPEIT2[20], we define the conditioning set as the $K$ haplotypes minimizing the Hamming distance with the current estimate $D = (h_1, h_2)$. In practice, this requires a quadratic scan of the haplotype data in $O(LN^2)$ that becomes prohibitive when facing large sample sizes (e.g., $N > 10,000$). In SHAPEIT3[5], we improved this by maintaining a clustering of the haplotypes so that Hamming distance minimization is only performed on the haplotypes (typically 4000) belonging to the same clusters than $D$. This approach exhibits close-to-linear running times in $O(LN \log N)$. In SHAPEIT4, we propose a fully linear approach in $O(LN)$ to assemble a conditioning set of $K$ haplotypes that share long matches with $D$. To do so, we build a PBWT of the full set of $2N$ haplotypes before each MCMC iteration (so 15 times total) by running the algorithm 2 from Durbin[19]. This algorithm is very fast and only requires a single $O(LN)$ sweep through the data. This gives prefix and divergence arrays that we store every eight variants by default to reduce memory usage. A prefix array is a permutation vector of the original haplotype indexes in which haplotypes sharing long prefixes (i.e., matches) are sorted next to each other. A divergence array specifies the starts of the matches. We extended the algorithm to also give a third array relating original haplotype indexes to permuted ones so that any haplotype can be located in the prefix or divergence arrays in constant time. Given these PBWT arrays, we can then update the haplotypes for $G$ as follows:

- We identify the $P$ haplotypes sharing the longest prefixes with $h_1$ and $h_2$ at each stored variants. By definition, these haplotypes are sorted next to $h_1$ and $h_2$ in the prefix arrays and the procedure has therefore running times proportional to $O(L)$.
- We collapse the $2LP/8$ resulting haplotypes into a list of $K$ distinct haplotypes (i.e., we remove duplicates). Long matches involve that the same haplotypes will be reported across multiple variants and therefore fewer distinct haplotypes once collapsed.
- We run the LSM onto the resulting conditioning set to get new estimates for $G$.

Using the same set of conditioning haplotypes across entire chromosomes is inefficient since some of them would only be informative at particular genomic regions. To account for this, we implemented the phasing procedure described above in a sliding window of size $W$ ($W = 2$ Mb by default) similarly to what has been already done in SHAPEIT2. Overall, this gives a procedure that has multiple interesting properties. First, it provides informative haplotypes for the LSM as they are guaranteed to share long matches with $D$ in the region of interest. Second, it is fast since building the PBWT and finding matches are done in running times proportional to $O(LN)$. Finally, it gives a number of conditioning haplotypes that varies depending on the matches found in the PBWT: longer matches involve fewer distinct haplotypes and therefore smaller $K$. This implies that the size of the LSM (i) adapts to the local level of relatedness between individuals, (ii) shrinks as the MCMC converges, and (iii) gets smaller for large sample sizes (i.e., sub-linearity with sample size).

**MCMC iteration scheme**. In SHAPEIT2 and 3, we start from a random haplotype resolution for all individuals and then perform 35 MCMC iterations to converge towards a reasonable haplotype resolution. In SHAPEIT4, we implemented three new features to improve the mixing of the MCMC:

- Initialization: We use a quick-and-dirty initialization of the haplotypes so that the MCMC does not start from a random resolution. To do so, we use a simplified version of the recursive phasing approach implemented in the PBWT software package (see https://github.com/richarddurbin/pbwt), which provides initial haplotype estimates very quickly in $O(NL)$ with SERs in the range of 8–10% in our data. In practice, we first build a vector $B_{L+1}$ that contains the $2N$ alleles carried by the $N$ individuals at variant $l+1$ in which reference alleles are encoded as $-1$, alternative ones as 1 and those occurring at heterozygous genotypes as 0. Then, we phase genotypes by progressively imputing missing data in $B_{L+1}$ (i.e., zeros) given the PBWT prefix array $A_l$ derived from the previous position $l$. Specifically, we make inference on the alleles carried by the two haplotypes of a given individual by copying those carried by neighbors as defined by the PBWT prefix array $A_l$ (i.e., located just before and after in $A_l$). Of note, for efficiency reasons, this procedure also requires to keep track of indexes of the haplotypes in $A_l$: we store those in a vector $I_l$ (see Fig. 5). Details on the exact procedure can be found in Fig. 5. Once all missing data filled, we build the prefix array $A_{l+1}$ for position $l+1$ and move to the next, $l+2$, until completion.
- IBD2 protection: We designed an approach to prevent SHAPEIT4 to copy haplotypes across individuals sharing both of their haplotypes IBD (i.e. IBD2). This typically happens in the case of siblings and constitutes a converge trap that can really hurt accuracy (the two individuals just copy

```
T <- 4
U <- 1
while T > 0 & U > 0
    U <- 0
    for i = 0 -> N do
        p_1 <- 0, n_1 <- 0, p_2 <- 0, n_2 <- 0
        a_1 <- I_1[2i]
        a_2 <- I_1[2i+1]
        if (a_1 > 0) p_1 <- B_{1+1}[A_1[a_1-1]]
        if (a_1 < 2N) n_1 <- B_{1+1}[A_1[a_1+1]]
        if (a_2 > 0) p_2 <- B_{1+1}[A_1[a_2-1]]
        if (a_2 < 2N) n_2 <- B_{1+1}[A_1[a_2+1]]
        S_1 <- p_1+n_1
        S_2 <- p_2+n_2
        if B_{1+1}[2i] = 0 & S_1 - S_2 = T then
            B_{1+1}[2i] = +1
            B_{1+1}[2i+1] = -1
            U <- 1
        if B_{1+1}[2i] = 0 & S_2 - S_1 = T then
            B_{1+1}[2i] = -1
            B_{1+1}[2i+1] = +1
            U <- 1
    if U = 0 then T <- T-1

Build prefix array A_{l+1} using algorithm 1 of ref. 19

    for h = 0 -> 2N do
        I_{1+1}[A_{1+1}[h]] = h
```

**Fig. 5** Phase genotypes in $B_{l+1}$ using prefix array $A_l$.

their haplotypes without making any updates). To do so, we extended the algorithm 3 of Durbin[19] to deal with the tri-allelic nature of genotype data and report genotype matches between individuals that are larger than $W$ Mb (i.e., the sliding window). We then use the reported matches to define local constraints that we account for when building the conditioning sets of haplotypes. By constraint, we mean here triplets $(i, j, w)$, where $i$ and $j$ are two individuals that are IBD2 in window $w$. In other words, when updating the phase of an individual in a given window, we prevent from copying other individuals that are IBD2 in this window.
- Specialized iterations: We designed three different types of iterations to help MCMC mixing. A burn-in iteration ($b$) uses transition probabilities in the genotype graphs to sample new pairs of haplotypes. A pruning iteration ($p$) uses transition probabilities for sampling and also for trimming unlikely haplotype configurations in the genotype graphs by merging consecutive segments. Of note, this pruning iteration differs from previous SHAPEIT versions: instead of doing a single pruning stage made of multiple iterations ($=8$), we perform multiple stages of pruning (by default 3), each made of a single iteration. Finally, a main iteration ($m$) samples haplotypes and stores transition probabilities so that they can be averaged at the end of the run to produce final estimates. The user can specify any sequence of iterations and the default is $5b, 1p, 1b, 1p, 1b, 1p, 5m$, which we found to perform well.

As a consequence of these three features, SHAPEIT4 needs a small number of MCMC iterations to reach good level of convergence: by default, it only performs 15 iterations as opposed to previous versions that required 35 iterations (2.3× more iterations).

**Reference panel**. SHAPEIT4 can borrow information from large reference panel of haplotypes, which proves to be particularly useful when phasing only few individuals. To achieve this, SHAPEIT4 simply considers the additional reference haplotypes when building or updating the PBWT to allow conditioning haplotypes to also originate from the reference panel. Besides this, the iteration scheme remains remarkably identical to the algorithm described above. Of note, SHAPEIT4 only retains variants in the overlap between the main and the reference panel.

**Haplotype scaffold**. SHAPEIT4 also allows for some heterozygous genotypes to be phased a priori. This approach has been previously introduced in SHAPEIT2 in the context of the 1000 Genomes project to perform genotype calling from low

coverage sequencing data[20]. This scaffold of pre-phased heterozygous genotypes can originate for instance from pedigrees in which many of the children' genotypes can be accurately phased using Mendel inheritance logic. In practice, we implemented this functionality by simply pruning out all haplotype configurations in the genotype graphs that are inconsistent with the available scaffold of haplotypes (Supplementary Fig. 9). Of note, SHAPEIT4 does not have any requirements in terms of variant overlap between the main genotype data and the scaffold haplotype data and therefore allows the latter to be derived from SNP array data while the former from high-coverage sequencing data. In the context of this work, we demonstrate the potential of this approach by deriving scaffolds from either trios or massive reference panels.

**Phase informative reads.** SHAPEIT4 allows haplotype estimation to be informed by sequencing reads overlapping multiple heterozygous genotypes. Haplotype assembly methods such as Whatshap[12] or HapCut[13] are very efficient at regrouping heterozygous genotypes within phase sets: groups of nearby genotypes for which the phase is inferred from sequencing reads that overlap them. In practice, SHAPEIT4 accommodates phase sets in a probabilistic manner (Supplementary Fig. 9). When sampling new haplotypes for $G$, it assumes an error model in which paths in the genotype graphs that are consistent with the phase sets receive more weight than paths that are not. In other words, SHAPEIT4 samples haplotypes from the distribution $P(D|H, R) \propto P(D|H)P(D|R)$, where $R$ represents the available phase sets for $G$ and assuming that $H$ and $R$ are conditionally independent of $D$. The sampling strategy is essentially the same than the one we previously developed for SHAPEIT2[17]. Only the way we compute the distribution $P(D|R)$ changes. We now use a simpler version in order to easily accommodate with new developments in sequencing technologies and haplotype assembly methods. Briefly, the distribution $P(D|R)$ is now controlled by a parameter that defines the expected error rate in the phase sets and not directly computed from the sequencing reads. By default, we assume an error rate of 0.0001 and used this value in all reported results. As a consequence, the sampled haplotypes $D$ for $G$ are not necessarily consistent with all the phase sets which allows correcting phasing errors in sequencing reads when phase sets are too discordant with population-based phasing. Of note, this method comes with three additional properties as a result of the data structure we used to store the space of possible haplotypes. First, the method can handle cases in which phase sets are disjoint (i.e., overlapping), which typically occurs in case of paired-end reads with large insert sizes. Second, the method does not use phase sets that are only informative for non-consecutive segments in the genotype graph structure, which occurs for a phase set that connects two distal heterozygous genotypes spanning multiple other heterozygous genotypes not included in the phase set. This may happen in the case of paired-end reads with very large insert sizes or in the case of Hi-C data for instance. As a segment of the genotype graph initially contains three heterozygous genotypes, SHAPEIT4 cannot use phase sets spanning more than three heterozygous genotypes in the first iterations. However, since consecutive segments are merged in the pruning iterations, SHAPEIT4 is still able to use phase sets spanning more than three heterozygous genotypes in the last set of iterations. Third, when using phase sets together with a haplotype scaffold, the phase given by the scaffold is always prioritized which means that reads inconsistent with the scaffold are automatically discarded from the analysis.

**I/O interface.** All the input and output interface implemented in SHAPEIT4 is built on the High-Throughput Sequencing library (HTSlib[30,31]) so that genotype and haplotype data is read and written in either variant call format (VCF) or its binary version; the BCF format. This has multiple benefits. All data management on input and output files can be done using standard tools such as bcftools[31]. Using the BCF format significantly speeds up I/O operations. VCF/BCF formats natively define phase sets, which facilitates the integration of phase information contained in sequencing reads in SHAPEIT4. HTSlib allows reading simultaneously multiple VCF/BCF files, which facilitates complex SHAPEIT4 runs combining, for instance, genotype data with phase sets, a reference panel of haplotypes and a scaffold of haplotypes.

**The UKB data sets.** We generated genotype data sets with large sample sizes from the full release of the UKB data set containing 488,377 individuals genotyped by SNP arrays at 805,426 genetic variants. To do so, we proceeded according to the five following steps. First, we filtered out all genetic variants with more than 5% missing genotypes. Second, we extracted data only for chromosome 20, resulting in 18,477 variants in total. Third, we build trio candidates (two parents and one child) from the pairwise kinship and IBS0 estimates (identity-by-state equals 0) between individuals that have been measured as part of the UKB study[2] and took the first 500 trios minimizing Mendel inconsistencies between parents and children genotypes. The 500 children constitute high-quality validation data since their haplotypes can be almost entirely resolved using Mendel inheritance logic. Fourth, we merged these 500 individuals with multiple random subsets of other UKB individuals in order to build 11 genotype data sets comprising between 500 and 400,000 individuals in total. Finally, we remove all individuals containing more than 5% missing genotypes in each data set. This gave us 11 data sets comprising 499, 995, 1997, 4973, 9939, 19,894, 49,752, 99,452, 198,894, 298,383, and 397,839

individuals for which genotype data is available at 18,477 variants. To mimic phasing from large reference panels, we used the best haplotype estimates obtained for each sample size from, which we removed the haplotypes of the 500 children. This resulted in 10 reference panels comprising 992, 2996, 8948, 18,880, 38,790, 98,506, 197,906, 396,790, 595,768, and 794,680 haplotypes defined at 18477 variants. All reference panels have been compressed to speed up I/O of the downstream phasing runs: we used BCF formats for both SHAPEIT4 and Eagle2 (using bcftools v1.4) and bref3 format for Beagle5 (using bref3.28Sep18.793.jar from https://faculty.washington.edu/browning/beagle/). For the main panels that were phased against these reference panels, we used the 500 trio children that we merged with 4500 and 49,500 other UKB samples. This resulted in three main panels with 500, 5000, and 50,000 individuals with genotype data at 18,477 variants. Of note, all data management was done using bcftools v.1.4[31].

**Phasing runs on the UKB data sets.** We phased the UKB data sets using SHAPEIT3 and three other widely used phasing methods: SHAPET3 (https://jmarchini.org/shapeit3/), Eagle v2.4 (https://github.com/poruloh/Eagle), and Beagle5 (beagle.14May18.ff7.jar; https://faculty.washington.edu/browning/beagle/beagle.html). All runs were done on a RedHat server with Intel(R) Xeon(R) CPU E7-8870 v4 @ 2.10 GHz (80 physical cores and 160 logical cores) and 3 Tb of RAM. Eagle v2.4 and Beagle5 have been run with default parameters. SHAPEIT3 has been run using the options–states 200,–cluster-size 4000 and–early-stopping. All methods were run using 10 threads to speed up computations. The reported running times and memory usages were obtained using the GNU time v1.9 command. The running times are computed as the sum of the User and System times and the memory usage as the Maximum resident set size. The error rates of each method was measured using the SER on the 500 trio children. Specifically, we enumerated all heterozygous genotypes (i.e., hets) in the 500 trio children that can be phased using Mendel inheritance logic (i.e., no triple hets and no Mendel inconsistencies). Then, we computed the SER as the fraction of successive pairs of hets that are incorrectly phased over all possible pairs. Confidence intervals for the SER are defined as binomial 95% confidence intervals and were computed using the R/binconf of the R/Hmisc package (https://cran.r-project.org/web/packages/Hmisc/index.html). For the analysis in which we introduced genotyping errors, we measured switch error rates only between genotypes without simulated errors.

**The GIAB data sets.** To assess performance of SHAPEIT4 on sequencing data, we use the high-quality phased genotype data generated for the NA12878 individual by the GIAB consortium[23]. In order to get population scale data, we merged the high-coverage genotype data of the GIAB with 502 unrelated European individuals sequenced as part of the phase 3 of the 1000 Genomes project (KGP3)[25]. To do so, we proceeded as follows. First, we only used variants on chromosome 20. Second, we assume NA12878 to be homozygous reference allele at all KGP3 variants not typed in GIAB and removed all GIAB variants not typed in KGP3. Third, since NA12878 has also been sequenced in KGP3, we only retained variants for which GIAB and KGP3 genotypes are concordant. Finally, we only kept variants that were phased by GIAB owing to some family data (i.e., multiple sequenced individuals from the NA12878 family). Of note, NA12878 was not included in the final set of 502 KGP samples we used for merging. In total, this procedure gave us a data set on chromosome 20 comprising genotype data for 503 European individuals across 507,181 genetic variants amongst which 478,581 are SNPs and the rest is a mixture of short indels and large SVs. Of note, all data management was done using bcftools v1.4[31]. The GIAB consortium has generated sequence data using multiple sequencing technologies for NA12878. In the context of this work, we used: (a) SOLiD5500W, (b) Complete Genomics, (c) Illumina HiSeq, (d) PacBio, and (e) 10x Genomics. Summary statistics for all this sequence data is given Table 1. We used the five sets of sequencing data to phase as much as possible of the NA12878 genotype data. In practice, we used WhatsHap v0.15[12] with default parameters to phase NA12878 using sequence data (a) to (d). The proportion of heterozygous genotypes being phased by each type of sequence data is shown in Table 1. For 10x Genomics sequence data, we proceeded quite differently since this technology relies on barcoded reads and not on long reads. In this case, we used the set of heterozygous genotypes that can be phased using the Long Ranger method provided by 10x Genomics. We have not run the method ourselves but instead used the Long Ranger outcome provided by the GIAB consortium. The outcome of WhatsHap or Long Ranger has been included in the VCF file as phase sets (i.e., field PS) so that SHAPEIT4 can use it.

**Scaffolding the GIAB data.** We generated two haplotype scaffolds for the data set described above either using family or large reference haplotype panel. For the first scaffold, we used genotype data derived from Illumina OMNI2.5M for a larger set of individuals containing multiple trios and duos. Out of the 503 individuals with sequence data, 190 of them (37.8%) are present in the OMNI data together with parents or children so that we could fix the phase at 6.59% of the heterozygous genotypes in the sequence data using Mendel inheritance logic. For the second scaffold, we phased the 16,805 variants in the overlap with UKB data on chromosome 20 using 795,678 UKB haplotypes as reference panel. This allowed fixing the phase at 6.25% of the heterozygous genotypes in the sequence data.

**Phasing runs on the GIAB data sets**. Using SHAPEIT4, we phased the 503 individuals with sequence data across all possible combinations of six types of sequencing reads (i.e., no reads, SOLiD5500W, Complete Genomics, Illumina HiSeq, PacBio, and 10x Genomics) and three scaffolds (no scaffold, OMNI, and UKB scaffolds). Each run was repeated five times using different random number generator seeds to assess variability of the results. Accuracy of phasing was measured using the switch error rate as described above on NA12878 using the phase provided by GIAB as reference. The phase here was determined from the Illumina Platinum Genomes extended family data (17 individuals sequenced at 50× from the pedigree 1463, the CEPH pedigree that includes NA12878 as grandchild). Of note, we also measure switch error rates between variants falling within identical phase set (see Fig. 4c), thereby ignoring the phase between variants belonging to distinct phase sets.

**Reporting summary**. Further information on research design is available in the Nature Research Reporting Summary linked to this article.

## Data availability

We used the full release of the UKB (http://www.ukbiobank.ac.uk/). This work was conducted under UKB project 35520. We used the following release of the GIAB (ftp://ftp-trace.ncbi.nlm.nih.gov/giab/) and this release of the phase 3 of the 1000 Genomes project (ftp://ftp.1000genomes.ebi.ac.uk). We downloaded this version of the larger set of 1000 Genomes samples genotyped on Illumina OMNI2.5M (ftp://ftp.1000genomes.ebi.ac.uk). The sequencing data we used for GIAB are available from this location (ftp://ftp-trace.ncbi.nlm.nih.gov/giab/ftp/data/NA12878/) and 10x Genomics VCF populated with PS field (ftp://ftp-trace.ncbi.nlm.nih.gov/giab/ftp/data/NA12878/10Xgenomics_ChromiumGenome_LongRanger2.1_09302016/NA12878_hg19/NA12878_hg19_phased_variants.vcf.gz). All other relevant data is available upon request.

## Code availability

SHAPEIT4 is available on the GitHub webpage (https://odelaneau.github.io/shapeit4/). The code used for computing switch error rates is also on github (https://github.com/odelaneau/switchError). All code is licenced under the MIT licence.

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

## Acknowledgements

This work was conducted under UK Biobank project 35520. O.D. is funded by a Swiss National Science Foundation (SNSF) project grant (PP00P3_176977). M.R.R. is funded by a Swiss National Science Foundation (SNSF) project grant (31003A-179380) and by core funding from the University of Lausanne. J.L.M. acknowledges funding for this work from the European Research Council (ERC; grant 617306). E.T.D. is funded by the Louis-Jeantet Foundation, European Research Council and Swiss National Science Foundation.

## Author contributions

O.D. designed the study, implemented the code, performed the experiments, and wrote the manuscript. O.D. conceived the original ideas thanks to fruitful discussions with J.L.M. and E.T.D., M.R.R. helped on the UKB data processing and the computations. J.L.M., J-F.Z., M.R.R., and E.T.D. discussed the results with O.D. and contributed to the final manuscript.

## Competing interests

E.T.D. is chairman and member of the Board, Hybridstat Ltd. O.D., J.-F.Z., M.R.R., and J.L.M. declare no competing interests.
