## [Peer Review File · Nature Communications]

Reviewers' comments:

Reviewer #1 (Remarks to the Author):

This paper describes a new version of the SHAPEIT software for phasing genotypes. The new version (SHAPEIT4) includes optimized code and a novel method for adapting the size of the model state space to the data to be phased. The new version also retains the method extensions from earlier versions for incorporating a reference panel, a scaffold of phased genotypes, and phase sets from phase-informative sequence reads. The method is evaluated using data from the UK Biobank (UKB) and Genome in a Bottle (GIAB). Phasing of the 400,000 UK Biobank participants show that SHAPEIT4 is 3.5x faster and has 7% lower error rate than the fastest, most accurate competing method. Accuracy results for GIAB when using phase information from sequence reads are also impressive.

The software has a simple user interface and is well-documented.

Comments and queries:

- Initialization algorithm. It would not be possible to implement the SHAPEIT4 initialization algorithm from the description that is provided. A complete description is needed. It's not clear what you mean by "one of two non-missing alleles on a surrounding haplotype" – are you referring to the nearest preceding and succeeding haplotype in the PBWT-sorted order? What happens if the alleles on the two haplotypes are not consistent with the masked heterozygote genotype? If a genotype at the marker being phased is an (unphased) heterozygote, how do you know the allele that is carried by a particular haplotype? How do you prioritize your choice of the source haplotypes from which to copy alleles? Why is an iterative update needed – why isn't one pass over the haplotype sufficient?
- The method of adapting the number of conditioning haplotypes to the data in each iteration is very elegant. Do you have any data showing how much accuracy is sacrificed when reducing the number of model states using this algorithm?
- What particular haplotypes are selected when there are more than P haplotypes sharing the longest prefixes with a haplotype
- What does "most accurate sets of haplotypes estimated for UKB" mean in the section on phasing from large reference panels?
- In lines 163-64, when the GIAB data is first mentioned, you should report the sample size ($n=1$).
- The (minimum) interval for distinct phase sets are disjoint in Figure 1C. Does the software permit the phase set intervals to overlap?
- What happens if a phase set is inconsistent with a haplotype scaffold?
- How can the relative running time be 0 in Figure 2C? Do you mean "percent change in running time"?
- Does the tilde mean "proportional to" in line 390?
- What is the overlap (if any) between the 2 Mb sliding windows?
- Are the 6.59% and 6.25% of heterozygous genotypes in the phased scaffold computed for the GIAB

individual or for the GIAB + KGP3 EUR data? In your phasing, did you also include scaffolds for the KGP3 European samples?

- What do you mean by “account for local constraints” in lines 344-346? Are you simply prohibiting a haplotype from an individual X from being used as a reference haplotype for individual Y if X and Y are IBD2 in a window?
- The GIAB datasets (lines 456-461). Does the fourth point logically occur before the 2nd point (i.e. infer homozygous ref genotypes before assessing concordance)?
- How did GIAB arrive at the phasing that is used as the truth?
- In Figure 4A and 4B, are the “No sequences” results from phasing the KGP3 EUR data by itself?
- In Figure 4B, what percentage of $MAF > 1\%$ variants are on the Omni 2.5M scaffold?
- In Figure 4C, is the switch error in each read computed by treating the variants in each phase set as a separate chromosome so that there are no pairs of heterozygotes that bridge two phase sets?
- In Figure 4C, the line segment for Hi-seq is much shorter than for the other sequencing platforms. Can you comment why this is the case?
- Supplementary Figure 5. Shouldn't the probabilities of transitions from a particular state sum to 1? Is there a reason that the third segment has 3 heterozygotes, rather than 2? The transition probabilities in the figure would be easier to understand if there were text describing how (some) specific entries of $P(D,R)$ are determined, and how $P(D|H,R)$ is computed from $P(D|H)$ and $P(D|R)$.

Reviewer #2 (Remarks to the Author):

The authors describe the Shapeit4 method/software that builds on previous shapeit versions for haplotype phasing. Shapeit4 uses the PBWT data structure to improve computational efficiency. The use of this data structure for haplotype phasing was first shown by Loh et al. 2016.

Overall, the method is well-presented, and is a clear advance over other methods. The main advantage is on speed (3-4X faster), with accuracy being very similar. A small speed-accuracy tradeoff is also shown. In addition, the shapeit4 method is shown to have lower switch error rates and lower run-times compared to other methods. It can also leverage haplotype information from sequence reads and reference panels for phasing. On balance, it is a useful addition to the suite of population based haplotyping tools.

Major comments:

=====

1. The running-time for SHAPEIT4 per genome decreases as the number of individuals is increased. This is an interesting result. The authors fit a sub-linear function to the running time in Supp Fig 2. Since they use the word sub-linear, it is worth clarifying is a bit, because you can get truly sub-linear only if you do not read the input genotypes data. Simply reading in the input data once would be linear, right? Perhaps, the running time of SHAPEIT4 can be modeled as:

$c1.N + c2.N^p$ where $p < 1$ and $c1$ and $c2$ are constants

For very large values of 'n', the first term can be larger than the second term. While this likely happens at very large values of 'N', it is not technically correct to say that SHAPEIT4 has sub-linear complexity. One could say that the running-time for SHAPEIT4 increases sub-linearly with N (for N in a certain range), and explain it a bit more.

2. SHAPEIT4 with P=4 outperforms Eagle2 and Beagle5 (Figure 5). Both Beagle5 and Eagle2 were run with default parameters. These programs also have options to improve accuracy by using larger 'K' (number of conditioning haplotypes), as shown in Loh et al. 2016. For a fair comparison, it would be good to consider these options.

3. On line 98, the authors state that "Then, we model the resulting phase sets as probabilistic constraints in the SHAPEIT4 haplotype sampling scheme so that 100 haplotype configurations consistent with them are favoured but not necessarily sampled. This is controlled by a parameter that defines the expected error rates in the phase sets (default is 0.0001)."

However, this does not take into the account the uncertainty in the phase information provided by the sequence reads. For variants with low coverage or conflicting evidence from sequence reads, the error of the phase set is greater. This should be discussed.

4. The authors previously described a method (Delaneau et al. AJHG 2013) to combine haplotype information from sequence reads with statistical phase information. Results were shown for long reads as well and the method is implemented in SHAPEIT2. The authors should discuss how the method implemented in SHAPEIT4 compares with this approach.

5. All results are shown on European ancestry individuals (UK Biobank and GIAB). African populations have higher levels of diversity and lower LD. It would be useful to analyze data from non-European ancestry populations such as those from the 1000 Genomes Project.

Minor comments:

=====

Supp Figure 4: PackBio -> PacBio

In Figure 2(C), it would be better to show the run-time per genome as a function of total number of samples.

line 213: the SHAPEIT4 -> SHAPEIT4

line 414: what does IBS0 refer to?

line 43: Hi-C based haplotyping allows for chromosome-scale phasing of genomes. Further, Porubsky et al. 2017 have shown that dense and chromosome-spanning haplotyping is feasible using sequence reads only. Therefore, this statement is not correct.

Installation:

=====

I was not able to install the software. I installed the Boost library but ran into an issue with the boost paths:

```
gmap_reader.cpp: (.text._ZN10input_fileC1ENSt7__cxx1112basic_stringIcSt11char_traitsIcESaIcEEEE[_ZN10input_fileC1ENSt7__cxx1112basic_stringIcSt11char_traitsIcESaIcEEEE]+0xf6b): undefined reference to `boost::iostreams::detail::bzip2_base::~bzip2_base()'
```

It would be useful for providing instructions for installing some of the libraries needed for compilation, e.g.

```
apt-get install libbz2-dev
```

Reviewer #3 (Remarks to the Author):

This manuscript describes a new haplotype phasing algorithm, SHAPEIT4, that provides two important advances over existing phasing methods: (1) increased speed and accuracy at very large sample sizes, and (2) support for integration of (probabilistic) phase information from other sources, e.g., sequencing reads or a pre-phased scaffold. Both of these improvements will be welcome additions to geneticists' toolkits. The authors should also be commended for releasing SHAPEIT4 as open-source software.

The algorithmic ideas used by SHAPEIT4 are sensible, and as the authors demonstrate, using PBWT to select copying haplotypes enables sub-linear run time scaling as sample sizes increase. The benchmarks included in the manuscript are quite comprehensive and convincing. However, I do have a few comments and questions I would like to see the authors address in a revision.

1. I am curious whether the choice of restricting the reference set to the longest P matches at each position makes SHAPEIT4 any less robust to genotyping error. If I understand the Methods correctly, the authors chose to benchmark phasing methods on the 500 trio children minimizing Mendel errors between parents and children. The UK Biobank data set contains ~1,000 trios, so this benchmark corresponds to assessing performance on the "cleaner half" of the data. I would like to see how the

various methods compare when accuracy is stratified by genotyping quality (e.g., restricting the accuracy benchmark to the 100 best / 100 worst / 100 middle trios out of the ~1,000 total trios, where "best"/"worst" could be measured based on either Mendel errors or missingness rate).

2. Similarly, I would like to see how the various methods compare when accuracy is stratified by ancestry (either self-reported or PCA-based would be fine).

3. The main text of the manuscript is vague about the memory use of SHAPEIT4: "In terms of memory usage, we also find SHAPEIT4 to substantially improve upon SHAPEIT3 (3.5x decrease) and to offer performance comparable to Beagle5 and Eagle2 (Supplementary Figure 3)." Please report the memory use (absolute or relative) of SHAPEIT4 P=4, Beagle5, and Eagle2 on the N=400K data set in the main text, just as the main text reports phasing accuracy and running times for these methods on the N=400K data set.

4. The relative accuracy of methods at N=10K and N=20K is very hard to distinguish in Fig. 2A and Fig. 3A-C. Please use a log scale on the x-axis to make the left side of the plot more readable.

5. In the main text, when introducing the results on "Phasing large scale datasets" please provide a bit more detail about the benchmark data (which chromosome was phased, how many variants were analyzed, how 500 trios were chosen, and any particularly pertinent QC). I understand that this information is provided in the Methods, but a rough sense of the features of the benchmark data would be helpful to include in the main text.

Detailed responses to individual comments can be found below.

Reviewer #1 (Remarks to the Author):

This paper describes a new version of the SHAPEIT software for phasing genotypes. The new version (SHAPEIT4) includes optimized code and a novel method for adapting the size of the model state space to the data to be phased. The new version also retains the method extensions from earlier versions for incorporating a reference panel, a scaffold of phased genotypes, and phase sets from phase-informative sequence reads. The method is evaluated using data from the UK Biobank (UKB) and Genome in a Bottle (GIAB). Phasing of the 400,000 UK Biobank participants show that SHAPEIT4 is 3.5x faster and has 7% lower error rate than the fastest, most accurate competing method. Accuracy results for GIAB when using phase information from sequence reads are also impressive.

The software has a simple user interface and is well-documented.

We thank the reviewer for the positive feedback.

Comments and queries:

- Initialization algorithm. It would not be possible to implement the SHAPEIT4 initialization algorithm from the description that is provided. A complete description is needed. It's not clear what you mean by "one of two non-missing alleles on a surrounding haplotype" – are you referring to the nearest preceding and succeeding haplotype in the PBWT-sorted order? What happens if the alleles on the two haplotypes are not consistent with the masked heterozygote genotype? If a genotype at the marker being phased is an (unphased) heterozygote, how do you know the allele that is carried by a particular haplotype? How do you prioritize your choice of the source haplotypes from which to copy alleles? Why is an iterative update needed – why isn't one pass over the haplotype sufficient?

We edited this part of the manuscript to add more details and the pseudo code of the algorithm we used for initializing the haplotype resolutions.

- The method of adapting the number of conditioning haplotypes to the data in each iteration is very elegant. Do you have any data showing how much accuracy is sacrificed when reducing the number of model states using this algorithm?

We do not have any data comparing our algorithm to select conditioning states to the full model (i.e. containing all possible $2N-2$ states) as the latter is difficult to run in reasonable times in realistic situations. However, for the current set of results, we can already make two interesting observations:

1. PBWT based selection seems to select better conditioning states than the Hamming distance as used in previous version of SHAPEIT. For instance, SHAPEIT4-P=2 uses systematically $K < 100$ across all tested sample sizes (see sup. fig. 2E) and performs better than SHAPEIT3 using a fixed number $K=200$ conditioning states chosen to minimize Hamming distance.
2. Increasing the value of P in PBWT selection brings substantial accuracy improvements at a small computational cost compared to increasing the value of K with other methods (see figure 2D). Reaching accuracy levels similar to SHAPEIT4 with other methods would require very high values for K and therefore would be prohibitive in terms of running times.

All this suggests that even though some loss of accuracy is expected when selecting few conditioning states using PBWT, our approach seems to be way more efficient than the selection approaches used in other methods.

- What particular haplotypes are selected when there are more than P haplotypes sharing the longest prefixes with a haplotype

We select the first P haplotypes that are sorted next to the current estimate and discard the remaining ones. This arbitrary selection should not harm accuracy as all additional haplotypes sharing the same prefixes can be considered as redundant pieces of information for phasing.

- What does “most accurate sets of haplotypes estimated for UKB” mean in the section on phasing from large reference panels?

In this second benchmark, we used as reference panels the best haplotype estimates we get as part of the first benchmark. Since, multiple estimates were obtained depending on the method and parameters used, we simply selected those with the smallest switch error rates as measured as part of the first benchmark.

- In lines 163-64, when the GIAB data is first mentioned, you should report the sample size ($n=1$).

We added this information in the main text.

- The (minimum) interval for distinct phase sets are disjoint in Figure 1C. Does the software permit the phase set intervals to overlap?

Phase sets may overlap in terms of the genomic regions they define and SHAPEIT4 is able to handle this information. This may happen in the case of paired-end reads with large insert sizes for instance. However, this comes with some limitations that we now discuss in the relevant method section.

- What happens if a phase set is inconsistent with a haplotype scaffold?

Any read inconsistent with the scaffold is discarded from the analysis. We therefore assume the scaffold to be of higher quality than sequencing reads which, to us, is a reasonable assumption

when the scaffolds are derived from family information or phasing from very large reference panels. We now discuss this in the relevant method section.

- How can the relative running time be 0 in Figure 2C? Do you mean “percent change in running time”?

We thank the reviewer for spotting this mistake. We indeed meant “percent change in running time” and changed the figure accordingly.

- Does the tilde mean “proportional to” in line 390?

Yes, we meant proportional there. We edited the text to include the correct notation.

- What is the overlap (if any) between the 2 Mb sliding windows?

There is overlap and its size depends on the size of the segments used in the genotype graphs (recall that these segments initially contain 3 unphased heterozygotes and get bigger as we merge them in the pruning stage of the algorithm). In practice, the overlap between two successive windows corresponds to one segment in the genotype graph. In other words, the windowing varies from one sample to the other and is constrained by the segmentation done for getting the genotype graphs. Of note, the windowing for an individual also varies from one MCMC iteration to the other in order to prevent loss of accuracy near overlapping regions.

- Are the 6.59% and 6.25% of heterozygous genotypes in the phased scaffold computed for the GIAB individual or for the GIAB + KGP3 EUR data? In your phasing, did you also include scaffolds for the KGP3 European samples?

These percentages are given for the whole set of 503 KGP3 individuals. In practice, we tried to build scaffolds for the largest possible set of KGP3 samples and managed to get a pedigree derived scaffold for 39% of them and a UKB derived scaffold for 100% of them. We edited the text to make this clearer to the reader.

- What do you mean by “account for local constraints” in lines 344-346? Are you simply prohibiting a haplotype from an individual X from being used as a reference haplotype for individual Y if X and Y are IBD2 in a window?

This is exactly what we do. In a given window, we prevent individuals that are locally IBD2 to copy from each other. We identified these problematic regions using a PBWT pass designed for genotype data that is therefore able to find long matches between genotype vectors. We edited the manuscript to better describe the procedure we use.

- The GIAB datasets (lines 456-461). Does the fourth point logically occur before the 2nd point (i.e. infer homozygous ref genotypes before assessing concordance)?

The reviewer is correct. We indeed first imputed missing data before proceeding with the concordance based filtering. We corrected the manuscript accordingly.

- How did GIAB arrive at the phasing that is used as the truth?

We looked at accuracy only at variants for which the phase was given by the GIAB consortium and originally determined from the Illumina Platinum Genomes extended family data (17 individuals sequenced at 50x from the pedigree 1463, the CEPH pedigree that includes NA12878 as grand-child). We now give this information in the manuscript.

- In Figure 4A and 4B, are the “No sequences” results from phasing the KGP3 EUR data by itself?

The reviewer is correct. Only genotype data was used in these runs. We re-labeled the plots.

- In Figure 4B, what percentage of $MAF > 1\%$ variants are on the Omni 2.5M scaffold?

We find that 19.12% of the variants in the KGP+EUR data set at $MAF \geq 1\%$ are also typed in the Illumina OMNI 2.5M SNP array.

- In Figure 4C, is the switch error in each read computed by treating the variants in each phase set as a separate chromosome so that there are no pairs of heterozygotes that bridge two phase sets?

We indeed treated each phase set independently. We edited the manuscript to describe the procedure and also feel we should add a link to the github repository containing the code we use for computing switch error rates.

- In Figure 4C, the line segment for Hi-seq is much shorter than for the other sequencing platforms. Can you comment why this is the case?

This happens because after the haplotype assembly pass made from Hi-seq, there is almost no more room for improvement: the phasing is partial (57.9% of the hets) but of extremely high accuracy ($< 0.1\%$ error rate). We think that this high accuracy results from the high base qualities inherent to Hi-seq data.

- Supplementary Figure 5. Shouldn't the probabilities of transitions from a particular state sum to 1? Is there a reason that the third segment has 3 heterozygotes, rather than 2? The transition probabilities in the figure would be easier to understand if there were text describing how (some) specific entries of $P(D,R)$ are determined, and how $P(D|H,R)$ is computed from $P(D|H)$ and $P(D|R)$.

The reviewer is correct and transition probabilities should sum up to one. Showing here a complete and correct example with numbers is rather complicated. We therefore attempted to simplify the figure and guided the readers to the references with all required technical details.

Reviewer #2 (Remarks to the Author):

The authors describe the Shapeit4 method/software that builds on previous shapeit versions for haplotype phasing. Shapeit4 uses the PBWT data structure to improve computational efficiency. The use of this data structure for haplotype phasing was first shown by Loh et al. 2016.

Overall, the method is well-presented, and is a clear advance over other methods. The main advantage is on speed (3-4X faster), with accuracy being very similar. A small speed-accuracy tradeoff is also shown. In addition, the shapeit4 method is shown to have lower switch error rates and lower run-times compared to other methods. It can also leverage haplotype information from

sequence reads and reference panels for phasing. On balance, it is a useful addition to the suite of population based haplotyping tools.

We thank the reviewer for the positive feedback and agree that PBWT in the context of phasing was pioneered by Loh et al, 2016, even though the use we make of it is fundamentally different. We edited the text to make this more explicit in the discussion.

Major comments:

=====

1. The running-time for SHAPEIT4 per genome decreases as the number of individuals is increased. This is an interesting result. The authors fit a sub-linear function to the running time in Supp Fig 2. Since they use the word sub-linear, it is worth clarifying is a bit, because you can get truly sub-linear only if you do not read the input genotypes data. Simply reading in the input data once would be linear, right? Perhaps, the running time of SHAPEIT4 can be modeled as:

$c_1.N + c_2.N^p$ where $p < 1$ and c_1 and c_2 are constants

For very large values of 'n', the first term can be larger than the second term. While this likely happens at very large values of 'N', it is not technically correct to say that SHAPEIT4 has sub-linear complexity. One could say that the running-time for SHAPEIT4 increases sub-linearly with N (for N in a certain range), and explain it a bit more.

The reviewer is totally correct. We therefore followed his advice and fitted the suggested model on the SHAPEIT4 running times. This shows that SHAPEIT4 indeed mixes two components with different complexities:

- (a) The I/O operations and the PBWT passes that are linear with N.
- (b) The HMM computations that are sub-linear with N.

Assuming that $T = aN + bN^c$ where T and N stand for the running times and sample sizes, we estimated from the data the most likely values for the a, b and c coefficients (see new supplementary figure 1). From this, we find that $a \ll b$ for all tested values of P, meaning that the linear component starts dominating the running times only when N is greater than:

$$N > e^{\left(\frac{\log a - \log b}{c-1}\right)}$$

In the worst case scenario (SHAPEIT-P=1), we predicted this to happen when $N > 10^{10}$ which is far more than any sample sizes currently or soon available. This confirms that while the method is not sub-linear from a theoretical point of view, it scales sub-linearly in practice for all possible data sets. We changed the title of the manuscript and edited the discussion to make this point clear.

2. SHAPEIT4 with P=4 outperforms Eagle2 and Beagle5 (Figure 5). Both Beagle5 and Eagle2 were run with default parameters. These programs also have options to improve accuracy by using larger 'K' (number of conditioning haplotypes), as shown in Loh et al. 2016. For a fair comparison, it would be good to consider these options.

Changing the number K of conditioning states allows tuning the speed-accuracy trade-offs of the methods. We agree with the reviewer that this should definitely be shown for a fair comparison, even though we showed results for SHAPEIT4 at most with its default value for P (=4). Therefore, we modified this parameter 'K' in other methods using 3 configurations in 100K UKB samples: (i) 'fast' in which we halve default K, (ii) 'standard' in which we used default K and (iii) 'accurate' in which we double default K (**figure 2D**). This showed that (i) SHAPEIT4 provides the best trade-off

between speed and accuracy and (ii) that other methods would need prohibitive running times to reach the accuracy delivered by SHAPEIT4.

3. On line 98, the authors state that "Then, we model the resulting phase sets as probabilistic constraints in the SHAPEIT4 haplotype sampling scheme so that haplotype configurations consistent with them are favoured but not necessarily sampled. This is controlled by a parameter that defines the expected error rates in the phase sets (default is 0.0001)."

However, this does not take into the account the uncertainty in the phase information provided by the sequence reads. For variants with low coverage or conflicting evidence from sequence reads, the error of the phase set is greater. This should be discussed.

Our new model accounts for the uncertainty in the phase information with a simple model that assumes uniform certainty across the genome. This is however true that certainty in phasing may vary from one region to the other depending on the coverage or the mapping quality for instance. However, we believe this to have small impact for the following reasons. First, even by assuming a constant error rate, our method is able to correct for many mistakes in the phase sets, many of those likely to happen at uncertain calls. For instance, we divide by ~three the number of phasing errors in PacBio (figure 4C). In addition, we also re-run the entire benchmark using an expected error rate of 1% (so 100 times the default) and did not find big difference (**supplementary figure 6**).

4. The authors previously described a method (Delaneau et al. AJHG 2013) to combine haplotype information from sequence reads with statistical phase information. Results were shown for long reads as well and the method is implemented in SHAPEIT2. The authors should discuss how the method implemented in SHAPEIT4 compares with this approach.

As requested by the reviewer, we now provide some comparisons with our previous implementation when it was possible to run it (**supplementary figure 7**) and compare both conceptually in the method section. Our new results show that SHAPEIT4 is able to leverage phase sets at least as efficiently as SHAPEIT2 does and in much more flexible way.

We actually moved from the SHAPEIT2 model to a simpler version as implemented in SHAPEIT4 mostly to account for constant developments made in sequencing technologies and haplotype assembly. Any new type of sequencing data is different by nature and comes with a new error mode. This requires lots of development efforts in order to adapt the SHAPEIT2 model for new types of data. This explains why, we could not run our previous implementation on some data types. On the contrary, having a model based on phase sets as defined by the VCF specifications allows for easy integration with any new or existing sequencing data and haplotype assembly method. Overall, we took a pragmatic decision here and opted for versatility.

5. All results are shown on European ancestry individuals (UK Biobank and GIAB). African populations have higher levels of diversity and lower LD. It would be useful to analyze data from non-European ancestry populations such as those from the 1000 Genomes Project.

We followed the advice of the reviewer and now present some results on another collection of samples across a wide range of ancestries (**supplementary figure 3**).

Minor comments:

=====

Supp Figure 4: PackBio -> PacBio

This has been corrected.

In Figure 2(C), it would be better to show the run-time per genome as a function of total number of samples.

As suggested by reviewer 1, we re-labeled the y-axis to make the figure easier to understand.

line 213: the SHAPEIT4 -> SHAPEIT4

This has been corrected.

line 414: what does IBS0 refer to?

IBS0 stands for identity-by-state equals to 0, i.e. which is equal to 1 in the case of opposite genotypes (Ref/Ref versus Alt/Alt) and 0 otherwise.

line 43: Hi-C based haplotyping allows for chromosome-scale phasing of genomes. Further, Porubsky et al. 2017 have shown that dense and chromosome-spanning haplotyping is feasible using sequence reads only. Therefore, this statement is not correct.

We edited the introduction to account for this comment and included the references.

Installation:

=====

I was not able to install the software. I installed the Boost library but ran into an issue with the boost paths:

```
gmap_reader.cpp:(.text._ZN10input_fileC1ENSt7__cxx112basic_stringIcSt11char_traitsIcESalcEEE[_ZN10input_fileC1ENSt7__cxx112basic_stringIcSt11char_traitsIcESalcEEE]+0xf6b): undefined reference to `boost::iostreams::detail::bzip2_base::~~bzip2_base()'
```

It would be useful for providing instructions for installing some of the libraries needed for compilation, e.g.

```
apt-get install libbz2-dev
```

This issue relates to the installation of boost which is not really related to the installation of SHAPEIT4. On the webpage, we give some instructions to compile the source and it is difficult for us to predict all installation issues that users will encounter when compiling the source. As SHAPEIT4 relies on standard libraries (boost, etc ...), we are now working on making the source released together with an installation script based on autotools (e.g. configure/make/make install). We will release the installation script before any publication is made.

Reviewer #3 (Remarks to the Author):

This manuscript describes a new haplotype phasing algorithm, SHAPEIT4, that provides two important advances over existing phasing methods: (1) increased speed and accuracy at very large sample sizes, and (2) support for integration of (probabilistic) phase information from other sources, e.g., sequencing reads or a pre-phased scaffold. Both of these improvements will be welcome additions to geneticists' toolkits. The authors should also be commended for releasing SHAPEIT4 as open-source software.

The algorithmic ideas used by SHAPEIT4 are sensible, and as the authors demonstrate, using PBWT to select copying haplotypes enables sub-linear run time scaling as sample sizes increase. The benchmarks included in the manuscript are quite comprehensive and convincing. However, I do have a few comments and questions I would like to see the authors address in a revision.

We thank the reviewer for the positive feedback.

1. I am curious whether the choice of restricting the reference set to the longest P matches at each position makes SHAPEIT4 any less robust to genotyping error. If I understand the Methods correctly, the authors chose to benchmark phasing methods on the 500 trio children minimizing Mendel errors between parents and children. The UK Biobank data set contains ~1,000 trios, so this benchmark corresponds to assessing performance on the "cleaner half" of the data. I would like to see how the various methods compare when accuracy is stratified by genotyping quality (e.g., restricting the accuracy benchmark to the 100 best / 100 worst / 100 middle trios out of the ~1,000 total trios, where "best"/"worst" could be measured based on either Mendel errors or missingness rate).

This is a good point we did not think about and to explicitly address it, we designed an additional analysis in which we artificially introduced increasing amounts of genotyping errors in the current validation data of 500 trios. This shows that the PBWT state selection is surprisingly robust to genotyping errors (**supplementary figure 4**) and behaves better than other methods when the genotyping error rate becomes high. Our explanation: genotyping errors involve shorter matches and therefore more conditioning states to get reliable estimates. Therefore, methods able to tailor the number of conditioning states to the data are expected to behave better than those using a fixed number K.

2. Similarly, I would like to see how the various methods compare when accuracy is stratified by ancestry (either self-reported or PCA-based would be fine).

As requested by reviewer 2, we now show accuracy on an additional dataset mixing multiple ancestries (**supplementary figure 3**) and find that SHAPEIT4 performs well across all tested ancestries. This constitutes another evidence of the robustness of selecting a variable number of conditioning states.

3. The main text of the manuscript is vague about the memory use of SHAPEIT4: "In terms of memory usage, we also find SHAPEIT4 to substantially improve upon SHAPEIT3 (3.5x decrease) and to offer performance comparable to Beagle5 and Eagle2 (Supplementary Figure 3)." Please report the memory use (absolute or relative) of SHAPEIT4 P=4, Beagle5, and Eagle2 on the N=400K data set in the main text, just as the main text reports phasing accuracy and running times for these methods on the N=400K data set.

We corrected the manuscript as suggested by the reviewer and explicitly mention memory usages.

4. The relative accuracy of methods at $N=10K$ and $N=20K$ is very hard to distinguish in Fig. 2A and Fig. 3A-C. Please use a log scale on the x-axis to make the left side of the plot more readable.

We changed the scale of the x-axis on all the relevant figures to make the plots more readable for the smaller sample sizes we tested.

5. In the main text, when introducing the results on "Phasing large scale datasets" please provide a bit more detail about the benchmark data (which chromosome was phased, how many variants were analyzed, how 500 trios were chosen, and any particularly pertinent QC). I understand that this information is provided in the Methods, but a rough sense of the features of the benchmark data would be helpful to include in the main text.

We added a bit more details on the data but left most of the QC details in the methods section as we feel this information does not constitute results but instead methods.

REVIEWERS' COMMENTS:

Reviewer #1 (Remarks to the Author):

The authors' have provided a thorough response to the reviewer comments. This paper and software represent a substantial advance in computational phasing methods.

The only suggestion that I have is to note that the I_I variable in "Algorithm 1" is the inverse of the A array. This is implied by the last loop in the algorithm, but until I noticed this I was quite confused.

Reviewer #2 (Remarks to the Author):

The authors have addressed all of my concerns satisfactorily, and I have no issues with the manuscript.

Reviewer #3 (Remarks to the Author):

The authors have comprehensively addressed my comments and improved their (already strong) manuscript.

One typo caught my eye: "Positional Burrow-Wheeler Transform" should be "Positional Burrows-Wheeler Transform" throughout.

Reviewer #1 (Remarks to the Author):

The authors' have provided a thorough response to the reviewer comments. This paper and software represent a substantial advance in computational phasing methods.

The only suggestion that I have is to note that the I_I variable in "Algorithm 1" is the inverse of the A array. This is implied by the last loop in the algorithm, but until I noticed this I was quite confused.

We corrected this.

Reviewer #2 (Remarks to the Author):

The authors have addressed all of my concerns satisfactorily, and I have no issues with the manuscript.

Reviewer #3 (Remarks to the Author):

The authors have comprehensively addressed my comments and improved their (already strong) manuscript.

One typo caught my eye: "Positional Burrow-Wheeler Transform" should be "Positional Burrows-Wheeler Transform" throughout.

We corrected these typos.